# Characterization of the external exposome and its contribution to the clinical respiratory and early biological effects in children: The PROMESA cohort study protocol

Diana Marín[1]*, Luz Yaneth Orozco[1,2], Diana María Narváez[3], Isabel Cristina Ortiz-Trujillo[1], Francisco José Molina[2], Carlos Daniel Ramos[2], Laura Rodriguez-Villamizar[4], Shrikant I. Bangdiwala[5,6], Olga Morales[7,8], Martha Cuellar[7,9], Luis Jorge Hernández[10], Enrique Antonio Henao[11], Verónica Lopera[11], Andrea Corredor[12], María Victoria Toro[13], Helena Groot[3], Milena Villamil-Osorio[14], Diego Alejandro Muñoz[15], Roberto Carlos Hincapié[13], Ferney Amaya[13], Ana Isabel Oviedo[13], Lucelly López[1], Ricardo Morales-Betancourt[16], Beatriz Elena Marín-Ochoa[17], Oscar Eduardo Sánchez-García[13], Juan Sebastián Marín[18], José Miguel Abad[18], Julio Cesar Toro[19], Eliana Pinzón[20], Juan José Builes[21], Zulma Vanessa Rueda[1,22]

**1** School of Medicine, Universidad Pontificia Bolivariana, Medellín, Colombia, **2** School of Engineering, Universidad de Antioquia, Medellín, Colombia, **3** Human Genetics Laboratory, Universidad de los Andes, Bogotá, Colombia, **4** Department of Public Health, School of Health, Universidad Industrial de Santander, Bucaramanga, Colombia, **5** Department of Health Research Methods, Evidence and Impact, McMaster University, Hamilton, Canada, **6** Statistics Department, Population Health Research Institute, McMaster University, Hamilton, Canada, **7** School of Medicine, Pediaciencias Group, Universidad de Antioquia, Noel Clinic, Medellín, Colombia, **8** Department of Pediatrics, Hospital San Vicente Fundación, Medellín, Colombia, **9** Department of Pediatrics, SOMER Clinic, Medellín, Colombia, **10** School of Medicine, Universidad de los Andes, Bogotá, Colombia, **11** Secretaría de Salud, Alcaldía de Medellín, Medellín, Colombia, **12** Department of Pediatrics, ONIROS Centro Especializado en Medicina Integral del Sueño, Bogotá, Colombia, **13** School of Engineering, Universidad Pontificia Bolivariana, Medellín, Colombia, **14** Department of Pediatrics, Fundación Hospital Pediátrico la Misericordia, Bogotá, Colombia, **15** Universidad Nacional de Colombia, Medellin, Colombia, **16** School of Engineering, Universidad de los Andes, Bogotá, Colombia, **17** School of Social Communications and Journalism, Universidad Pontificia Bolivariana, Medellín, Colombia, **18** Healthcare Company, SURA, Medellín, Colombia, **19** ATB SERVICE SAS, Medellín, Colombia, **20** Secretaria distrital de Salud, Alcaldia de Bogota, Bogota, Colombia, **21** GENES Laboratory, Medellín, Colombia, **22** Department of Medical Microbiology and Infectious Diseases, University of Manitoba, Winnipeg, Canada

* dianamarcela.marin@upb.edu.co

**Data Availability Statement:** This article does not report data and the data availability policy is not applicable to this article.

## Abstract

### Background

Air pollution contains a mixture of different pollutants from multiple sources. However, the interaction of these pollutants with other environmental exposures, as well as their harmful effects on children under five in tropical countries, is not well known.

### Objective

This study aims to characterize the external exposome (ambient and indoor exposures) and its contribution to clinical respiratory and early biological effects in children.

**Funding:** DM is the principal investigator. Grant number 121080763273, Contract 757-2018. This work was funded by the Colombian Ministry of Science, Technology and Innovation (https://minciencias.gov.co/). The funders had no role in study design, data collection and analysis, decision to publish, or preparation of the manuscript. JMA and JSM, coinvestigators. Healthcare Company, SURA (https://www.epssura.com/). The funder provided support in the form of salaries for authors [JMA, JSM], but did not have any additional role in the study design, data collection and analysis, decision to publish, or preparation of the manuscript. The specific roles of these authors are articulated in the 'author contributions' section." JJB, coinvestigador. GENES laboratory (https://laboratoriogenes.com/). The funder provided support in the form of salary for author JJB, but did not have any additional role in the study design, data collection and analysis, decision to publish, or preparation of the manuscript. The specific role of this author is articulated in the 'author contributions' section.

**Competing interests:** JCT is the owner and CEO of ATB SERVICE SAS. He does not have access to any children's information and the Environmental information is transmitted in real time and the data will be public. JMA and SM are employed by the Healthcare Company, SURA. JJB is employed by GENES laboratory. This does not alter our adherence to PLOS ONE policies on sharing data and materials. The other authors do not have conflicts of interest to declare

## Materials and methods

A cohort study will be conducted on children under five (n = 500) with a one-year follow-up. Enrolled children will be followed monthly (phone call) and at months 6 and 12 (in person) post-enrolment with upper and lower Acute Respiratory Infections (ARI) examinations, asthma development, asthma control, and genotoxic damage. The asthma diagnosis will be pediatric pulmonologist-based and a standardized protocol will be used. Exposure, effect, and susceptibility biomarkers will be measured on buccal cells samples. For environmental exposures $PM_{2.5}$ will be sampled, and questionnaires, geographic information, dispersion models and Land Use Regression models for $PM_{2.5}$ and $NO_2$ will be used. Different statistical methods that include Bayesian and machine learning techniques will be used for the ambient and indoor exposures-and outcomes. This study was approved by the ethics committee at Universidad Pontificia Bolivariana.

## Expected study outcomes/findings

To estimate i) The toxic effect of particulate matter transcending the approach based on pollutant concentration levels; ii) The risk of developing an upper and lower ARI, based on different exposure windows; iii) A baseline of early biological damage in children under five, and describe its progression after a one-year follow-up; and iv) How physical and chemical $PM_{2.5}$ characteristics influence toxicity and children's health.

## Introduction

Air pollution is the main environmental risk factor according to the 2019 Global Burden of Disease report for both adults and children [1]. In 2016, over 4.2 million people worldwide died from this risk factor and 543 000 deaths in children under 5 years were attributable to the joint effects of Ambient Air Pollution (AAP) and Household Air Pollution (HAP). Exposure to air pollution from fine particulate matter [particles with diameter $\leq$ 2.5 μm ($PM_{2.5}$)] can occur both indoors and outdoors, and its importance lies in the fact that it is not only the main pollutant, but also one of the most widely studied pollutants for its effects on health [2–5]. $PM_{2.5}$ can be referred to as "primary" when it is directly emitted by polluting sources, or "secondary" when it is formed by the nucleation of agglomerating small particles, which easily bind to other toxic compounds such as heavy metals, volatile organic compounds, Polycyclic Aromatic Hydrocarbons (PAH) [6, 7], and even viruses or bacteria [8].

Other air pollutants, such as carbon monoxide (CO), ozone ($O_3$), sulfur dioxide ($SO_2$) and nitrogen dioxide ($NO_2$) are also monitored due to their adverse health effects. Short- and long-term exposure to these pollutants during pregnancy, infancy and childhood particularly those found in Traffic-Related Air Pollution (TRAP), have been associated with incidence of asthma, asthma exacerbations and sensitization to both aero- and food allergens [9, 10], acute lower respiratory infections [11, 12], decreased lung functions [13], neurodevelopment [14], adverse birth outcomes, otitis media, childhood cancer [5] and some molecular markers such as accelerated epigenetic aging and telomere length shortening [15, 16]. Furthermore, there is evidence of harmful health effects even below the current safe level guidelines [17].

Exposure to air pollutants, including PAHs, has also been associated with early biological effects [18]. Several epidemiological studies have used the Comet (single-cell gel

electrophoresis) and Micronucleus (cytokinesis-block micronucleus cytome assay) assays to detect DNA damage from exposure to genotoxic agents [19, 20]; their application in human biomonitoring studies has been widely recognized [21, 22]. Although evidence in children is less common than in adults, some studies suggest that genotoxicity biomarkers (i.e., micronuclei frequency and DNA breaks) are higher in children who are exposed to elevated air-pollutant levels from traffic, industry, and biomass burning [19, 20, 23, 24]. Moreover, DNA damage is more common in children with asthma [25, 26], which increases their risk of developing cancer [27]. Although most studies focusing on the early biological effects from air-pollutant exposure in children have been conducted in Europe, the few South American studies conducted in Brazil were performed on a population of children over five [23, 28–30]. Still, prospective studies focused on assessing the variability of the association between different air pollutants and early genetic damage are even less frequent.

Compared with adults, children are more susceptible to air pollution effects, even at minor levels of exposure [31], which is mainly attributed to the fact that children are often exposed to different forms of air pollution starting in their intrauterine life (by transplacental transmission). In addition, children's detoxification and metabolism systems are underdeveloped, their lungs continue to develop until approximately the age of eight, they exhibit higher respiratory rates and airway surface areas in proportion to their body size, and they are more physically active outdoors [5, 32]. Furthermore, prenatal exposure to these pollutants is associated with adverse outcomes at birth [33], with some studies even suggesting lifelong effects [5, 34]. Similarly, the incidence of childhood respiratory infections (the second cause of death in children under five) [32, 35] and childhood asthma (the main chronic disease observed in this population) have also been associated with prenatal exposure to TRAP [10, 36, 37]. However, estimates of these associations are not available in low- and middle-income countries, despite their documented needs [38].

For a more complete assessment of the risk to which children are exposed, the global effects of pollutants and their interactions must be addressed. This implies a comprehensive approach in which different pollutants are assessed and in which exposure quantification is not only solely based on concentration levels, but also on the toxicity levels associated with specific emission sources, which vary significantly and independently from pollutant concentration levels [39, 40].

Moreover, there are health effects associated with specific air-pollutant sources [41, 42]; and other aspects, such as PM structure and composition, may also influence human health and vary according to atmospheric changes and meteorological conditions, as well as the sampling site [19, 39, 40].

An approach that has recently received increasing attention is the exposome [43–45], as this approach provides a better understanding of the role that environmental factors play in disease etiology and pathophysiology. The exposome is defined as all environmental exposures experienced by individuals since their conception and throughout their entire lifetime, as well as differences in internal, specific external, and general external exposures [43, 46]. The exposome captures the fluctuating dynamics of environmental exposures, the diversity in their sources, and their interactions. This approach also considers that the influence of these exposures on individual health could be mediated by body responses against these exposures [46].

Therefore, the PROMESA (as per its Spanish acronym meaning: Origin of particulate matter and its effect on children's health) study, aims to characterize the external exposome and its contribution to clinical respiratory and early biological effects in children under five in two Colombian cities. Within this context, this manuscript describes the research study, its execution processes, methodology, and expected results.

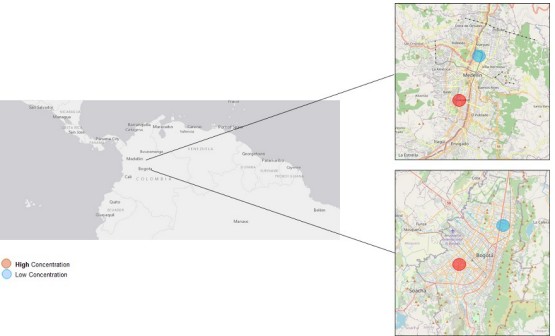

**Fig 1. Map of study zones.** Bogota and Medellin are the two biggest cities in Colombia. The zone with a high concentration of pollutants is indicated in red circle and the zone of low concentration in blue circle. Source: USGS National Map Viewer (http://viewer.nationalmap.gov/viewer/).

## Material and methods

### Study design

A prospective cohort of 500 children under five are to be followed for 1 year. Clinical respiratory and early biological effects as well as external exposome are to be assessed during this period.

### Setting

The PROMESA study will be conducted in Bogotá and Medellín, the two largest cities in Colombia, as both cities report, in their monitoring network records, that $PM_{2.5}$ is the pollutant that most deteriorates air quality (**Fig 1**). In the last five years, these two cities have experienced environmental emergencies due to $PM_{2.5}$ concentrations exceeding their maximum allowable daily levels (37 μg/m$^3$). In addition, in all monitoring stations in Medellín, annual average $PM_{2.5}$ concentrations have varied between 13 and 35 μg/m$^3$ [47]. In Bogotá, the reported annual variations have ranged between 9 and 36 μg/m$^3$ [48], also exceeding their maximum allowable annual levels (25 μg/m$^3$). Both cities are located in mountainous tropical zones. Medellín is located 1,500 meters above sea level in a narrow and semi-enclosed valley surrounded by mountains that reach heights of up to 2,800 meters above sea level, which favors atmospheric stability but causes air-pollutant stagnation. Bogotá is located at 2,650 meters above sea level in a high plateau within the Colombian Andes, and its eastern natural border is formed by hills that reach 3,650 meters above sea level, which sometimes favors the dispersion of air pollutants.

Our sampling sites were selected based on the following criteria:

1. Three different sources of air quality measurements identified high and low $PM_{2.5}$ levels in the same areas in 2017, 2018, and 2019

    a. High Concentration Zones: inventory of emissions above the 90th percentile, air quality monitoring stations with the highest number of days with a $PM_{2.5}$ Air Quality Index between 101 and 200. For these zones, the WRF, BRAMS-CAMx simulation model yields values exceeding 37 μg/m$^3$.

    b. Low Concentration Zones: inventory of emissions below the 10th percentile, air quality monitoring stations with the highest number of days with a $PM_{2.5}$ Air Quality Index between 0 and 50. For these zones, the WRF, BRAMS-CAMx simulation model yields values below 15 μg/m$^3$.

2. At least one healthcare provider from the Health Insurance Company must be operating within these high and low concentration zones.

3. At least one Air Quality Monitoring Station is operating within each zone.

## Study population

This study will randomly sample two-, three-, and four-year-old children affiliated to the Health Insurance Company who meet the following criteria:

1. The mother gave a singleton birth (the birth of only one child during a single delivery).

2. The child must live within 3 km of the air pollution and environmental parameter monitoring station that will be established in the selected area, in order to control the spatial variability of the air pollution emission sources monitored in each area [28].

3. The child's parent, caregiver, or legal representative must approve the child's participation in the study by signing the corresponding informed consent form.

This study will exclude preterm birth children (infants born alive before 37 weeks of gestation) as well as oxygen-dependent full-term children and children with cystic fibrosis, bronchopulmonary dysplasia treated with supplemental oxygen, congenital interstitial diseases, severe restrictive diseases, pulmonary hypertension, tracheo-bronchial malformations, obliterative bronchiolitis, and noncystic bronchiectasis. Children with other serious extrapulmonary conditions with pulmonary effects, such as neuromuscular diseases, lysosomal storage diseases, primary or secondary immunodeficiencies, congenital heart diseases, heart failure due to any cause, sickle cell anemia, autoimmunity and malignancy, shall also be excluded.

## Context

We will study the children of a single health insurance company in Colombia that has an insurance coverage of 94.3% in Medellín and 6.4% in Bogotá. Colombian health system has two types of health insurance: Contributive (people who have work or who pay by themselves [a self-employed person]) and Subsidized (those who cannot pay health insurance, and are subsidized the government as well as from resources from the contributive system). In addition, there are two structures: i) Health care insurance companies and, ii) Health care institutions that provide the clinical service. Health insurance companies can have clients from the contributive and subsidized regimens.

The distribution of clients from the health insurance company for both contributive and subsidized is similar to the general population and its clients live in all neighborhoods of both cities.

## Objectives and outcome measurements

PROMESA will not only establish a baseline for clinical and early biological effects in children, but also for their occurrence throughout the one-year of follow-up (**Table 1**). The primary objectives of this study are: (1) To estimate the prevalence and incidence of upper and lower Acute Respiratory Infections (ARI) in children under five, and (2) their association with prenatal and childhood external environmental exposures.

The secondary objectives are: (3) To estimate the prevalence of asthma and its control, exploring their relationship with prenatal and childhood external environmental exposures, (4) To determine the prevalence and incidence of genotoxic damage in children under five, and (5) its association with susceptibility genes and external environmental exposures in the

**Table 1. PROMESA endpoints and related outcomes.**

| |
|---|
| **Primary endpoints: ARI** |
| • Proportion of children with upper and lower Acute Respiratory Infections (ARI) at baseline |
| • ARI cumulative incidence and incidence density during the 12-month follow-up. |
| **Secondary endpoints: diagnosis of asthma, asthma control, DNA damage, gene mutation, mutagenicity** |
| • Proportion of children with diagnosis of asthma upon admission into the cohort. |
| • Proportion of children with asthma control during the 12-month follow-up |
| • Frequency of cells with micronucleus and nuclear buds at baseline, 6 and 12-month follow-up |
| • Frequency of binucleated, karyolitic, pycnotic, karyorrhetic and condensed chromatin cells at baseline, 6 and 12-month follow-up |
| • Proportion of DNA in tail at baseline, 6 and 12-month follow-up |
| • Proportion of children in each category of the damage index |
| • Proportion of BaP adducts upon admission into the cohort |
| • Number of revertants at baseline, 6 and 12-month follow-up |

BaP: Benzo[a]pyrene.

previous 0–7 days and (6) To characterize the nanostructure, morphology, chemical composition, and *in vitro* genotoxicity and mutagenicity of PM$_{2.5}$ in each city.

National guidelines will be followed for the clinical diagnosis of asthma [49]. This guide is based on the best international information available and necessary to make the diagnosis of asthma in patients under 5 years of age. A standardized protocol will be used to define the diagnosis and its severity, additionally, a validated TRACK questionnaire will be used for asthma control evaluation [50]. In **Table 2**, further information is provided about the timing and the information to be gathered, the buccal mucosa samples to be taken, and the environmental exposures being measured. The study duration is 2.5 years from July 2021 to January 2024

## Study procedures

Upon enrollment in the study, parents and caregivers will be interviewed using a tablet-aided version of a standardized questionnaire developed for HELIX [51, 52]. Questionnaire was translated into Spanish. The questionnaire includes information on children's diet, physical activity of the child, socioeconomic status, social capital of the family, stress of the mother, exposure to environmental tobacco smoke, cooking method at the home, cleaning products, bedroom location, noise perception, use of green spaces, commuting behavior, holidays and sun exposure.

Afterward, the children will be examined by a pediatric pulmonologist to confirm the diagnosis of asthma or upper or lower ARI, based on medical history and a general physical examination. A questionnaire was developed to evaluate this information and internal consistency was evaluated through consultation with four expert pediatric pulmonologists (S1 File).

The diagnostic criteria for asthma and its severity will be based on the national guidelines which is valid for children under 5 years of age [49]. The caregivers of children with asthma will be informed that the progress of the disease will be assessed every four months through the TRACK questionnaire [50]. In addition, the quality of life of the caregivers will be assessed every six months using the Pediatric Asthma Caregiver Quality of Life Questionnaire (PACQLQ) [53]. Both questionnaires have been properly validated for their use in Colombia. Caregivers will also be informed that if asthma or upper and lower ARI are detected, the child will be referred to the child's local healthcare provider. The monitoring of upper and lower

**Table 2. Scheduling of PROMESA study procedures and evaluations.**

| Procedures and Evaluations | Baseline | Monitoring* | | | | | | | | | | | Closeout |
|---|---|---|---|---|---|---|---|---|---|---|---|---|---|
| | 0 | $t_1$ | $t_2$ | $t_3$ | $t_4$ | $t_5$ | $t_6$ | $t_7$ | $t_8$ | $t_9$ | $t_{10}$ | $t_{11}$ | $t_{12}$ |
| **The child** | | | | | | | | | | | | | |
| Assess eligibility criteria | x | | | | | | | | | | | | |
| Informed consent | x | | | | | | | | | | | | |
| Entry questionnaire | x | | | | | | | | | | | | |
| Environmental exposure questionnaire [51, 52] | x | | | | | | x | | | | | | x |
| ARI Signs and symptoms questionnaire, visits to the emergency room and hospitalizations[1] | x | x | x | x | x | x | x | x | x | x | x | x | x |
| Clinical evaluation for the diagnosis of asthma and upper and lower ARI | x | | | | | | x | | | | | | x |
| TRACK Asthma control questionnaire [50] | x | | | | x | | | | x | | | | x |
| PACQLQ [53] | x | | | | | | x | | | | | | x |
| *Collection of Buccal Mucosa Cells* | x | | | | | | x | | | | | | x |
| Exposure biomarker | x | | | | | | | | | | | | |
| Early biological effect biomarkers | x | | | | | | x | | | | | | x |
| Susceptibility biomarkers | x | | | | | | | | | | | | |
| **The Environment** | | | | | | | | | | | | | |
| *PM$_{2.5}$ Collection through Sampling Campaigns* | x | | | | | | | | | | | | x |
| Chemical characterization of PM$_{2.5}$ | x | | | | | | | | | | | | x |
| Characterization of PM$_{2.5}$ morphology and nanostructure | x | | | | | | | | | | | | x |
| PM$_{2.5}$ *in vitro* genotoxicity and mutagenicity biomarkers | x | | | | | | | | | | | | x |
| Monitoring SO$_2$, O$_3$, NO$_2$, VOC, PM$_{2.5}$, and PM$_{10}$ | x | x | x | x | x | x | x | x | x | x | x | x | x |
| Wind speed/direction, temperature, and relative humidity measurements | x | x | x | x | x | x | x | x | x | x | x | x | x |

* t = time or month of child evaluation.

[1]The ICD-10 codes verified through medical history will be: Acute Lower Respiratory Disease (J00-J11.1), Pneumonia (J12.0-J18.9), Acute Bronchiolitis/Asthma (J21.0-J21.9/J44.8–46.0). ARI: Acute Respiratory Infection. PACQLQ: Pediatric Asthma Caregiver Quality of Life Questionnaire. TRACK: Test for Respiratory and Asthma Control in Kids.

ARI will be conducted through a monthly questionnaire, administered through phone conversations by a physician trained by the pediatric pulmonologists. The internal consistency was evaluated through consultation with expert pediatric pulmonologists (S1 File).

## Laboratory evaluations, assays, and handling

A buccal mucosa sample and a blood sample obtained by capillary puncture will be collected from all children every six months, for the assessment of early biological effects [alkaline Comet assay [54, 55] and micronucleus test [56]]. In addition, Benzo[a]pyrene (BaP) adducts in DNA will be determined as an exposure biomarker [57, 58], and the polymorphisms of five genes involved in DNA repair and in the metabolism of genotoxic agents (*CYP1A1*, *GSTM*, *P53*, *AHR* and *H2AX*) will be determined as susceptibility biomarkers. These genes were selected based on biological plausibility and limited to five because of budget constraints.

Both cities will use the same kits and a standardized protocol. All samples will be received, processed, and stored at the laboratories located in each city. The samples will be shipped from each clinical collection site to the corresponding laboratory, following standard sample-shipping procedures.

Furthermore, PM$_{2.5}$ samples will be collected at two different times of the year, based on the air pollutant concentrations. The two sampling campaigns will be conducted for two weeks in each sampling site (two sampling sites in each city) using high and low volume

samplers with Teflon and Quartz filters, based on the corresponding $PM_{2.5}$ assessments. After gravimetry, all filters will follow a chain of custody for later analysis at the laboratories.

$PM_{2.5}$ samples will be collected through pressurized hot water extraction (PHWE) and dispersive liquid–liquid microextraction [59]. The organic extracts obtained will be analyzed via gas chromatography–mass spectrometry (CG–MS) for 16 priority PAHs [60]. Likewise, transmission electron microscopy (TEM) and Raman spectrometry will be performed to determine the morphological and nanostructural characteristics [61]. Finally, the Kado version of the Ames test will be conducted [62] using *Salmonella typhimurium* TA100 and TA98 strains in the presence and absence of metabolic activation enzymes (±S9), and the alkaline Comet assay [55] and the micronucleus test [56] will be performed on the HepG2 cell line.

## Ethics and dissemination

All study procedures will be performed in accordance with the principles established by the World Medical Association Declaration of Helsinki and according to the ethical standards of Colombia. The protocol and informed consent were approved by the Health Research Ethics Committee of the Universidad Pontificia Bolivariana on April 18, 2018 (Approval N° 7–2018). Written informed consent will be obtained by trained study staff from all eligible children's caregivers that agree that her/his kid participates in the study prior to enrolment. According to Colombian law, written consent is not required from children under five, but the objectives of the study will be explained to them in an age-appropriate manner, and their concerns will be addressed. Those kids that refuse to participate in the study will not be included.

All information will be treated as confidential, wherein a unique code will be assigned to each participating child. This unique code will be used to identify each child in questionnaires, biological samples, and databases.

Study results will be disseminated in peer-reviewed journals, international conferences and local events with public health authorities together the community advisory. Study results will also be aggregated and included in multimedia educational and recreational materials distributed to the healthcare providers to which children are affiliated.

## Statistical considerations

**Sample size calculation.** Sample size was calculated in Epidat 3.1 using a level of significance of 0.05, an 80% power, a 37.5% incidence of upper and lower ARI in an area of high exposure to $PM_{2.5}$ or $NO_2$ [12], a 25.3% incidence in an area of low exposure to $PM_{2.5}$ or $NO_2$, and an expected Relative Risk (RR) for our study of 1.48 thus requiring a minimum of 228 children in each exposure zone (high and low concentration zones), for a total of 456 children. The sample size will be adjusted to 500 to cover for possible subject losses during follow-up. With this sample size, we will be able to evaluate genotoxic damage incidence differences by exposure zone at a power of 92%, and even simultaneously by city at a power of 80%. Secondary asthma outcomes and their control will be assessed through exploratory analysis because we were unable to expand the sample size due to budget constraints.

**Land Use Regression (LUR) model.** The annual average concentration of $PM_{2.5}$ and $NO_2$ at the child's home will be estimated using the LUR models developed by another research project conducted by Universidad Industrial de Santander in five Colombian cities. For constructing these models, the previously described methodology will be used [63]. In short, LUR models are based on the simultaneous measurement of air pollutants such as $PM_{2.5}$ and $NO_2$ in various places throughout urban areas. The variables derived from Geographic Information Systems, such as traffic density and volume, population density and land use, are used as predictor variables to model spatial concentration variations at specific sites and time periods.

The exposure history of each child will be constructed from the prenatal period to the age of the child upon entering the cohort. Therefore, parents will be required to provide information on their residential location history throughout the child's life period. Then, to account for seasonal pollutant variability, the LUR models will be adjusted for each exposure window (prenatal, one, two, three, and four years or age) based on the time series from the daily measurements provided by the air quality monitoring network in each city. In addition, if the family has moved to a different home once or more than once throughout this period, average annual concentrations will be estimated based on the time spent in each location [64, 65].

**Dispersion model.** As part of this study, small-scale dispersion models will allow us to dynamically predict $PM_{2.5}$ and $NO_2$ levels at a local level [66]. These models are based on conservation laws to simulate air-pollutant dispersions. Our model will use information from the LUR model and the PROMESA monitoring stations as input data to monitor $SO_2$, $O_3$, $NO_2$, VOC, $PM_{2.5}$, and $PM_{10}$ in each exposure zone and throughout the study, as well as meteorological conditions, wind speed and direction, ambient temperature, and relative humidity. The monitoring station is an electronic system composed of sensors to measure air-pollutant compounds. Measurements from these sensors will be validated against the records provided by the monitoring networks in each city.

**Dynamic lung model.** Based on the physiology, anatomy, and operation of the respiratory system, a phenomenological-based semi-physical model will be constructed. In this model, environmental exposure variables will be correlated against the outcomes studied to predict the risk to develop outcomes as the initial conditions change. In addition, respiratory system models will be used to complement the construction of our Dynamic Lung Model. We want to point out that the dispersion model's predictions will be used as inlet conditions for the dynamic lung model.

**Statistical analyses.** To characterize prenatal and postnatal environmental exposures, we will conduct descriptive analysis to identify correlations between exposure groups (air pollutants, $PM_{2.5}$, surrounding natural spaces, meteorology, traffic, tobacco smoking, lifestyle, and socio-economic capital) and their individual exposures. To determine exposome-health outcomes associations we will use the incidence of any clinical primary outcome and environmental exposures as predictors. The selected models will be developed considering the first and recurring episodes as outcomes. These models will contemplate the temporal and spatial correlation between environmental exposures adjusted for age, sampling site, ancestry percentages, and susceptibility biomarkers. Furthermore, additional confounding and interaction variables will be assessed based on the reported literature. The statistical methods that will be used to study the associations are Deletion–Substitution–Addition (DSA) algorithm, Bayesian Kernel Machine regression, and Weighted Quantile Sum Regression [67]. The Elastic Net (ENET) and Least Absolute Shrinkage and Selection Operator (LASSO) will also be used to deal with repeated measures but this is an area of research still under development. Sensitivity analyzes including alternative models will also be conducted to assess different assumptions about air-pollutant concentration distributions, correlation heterogeneity, unmeasured confounding and potential biases. For assessing secondary outcomes, the only difference will be their responses, as they will include average genotoxic and mutagenic damage changes These secondary outcomes will be adjusted using the same variables as the main assessments, but will also be adjusted for the BaP exposure biomarker. In addition, the relationship between the secondary outcomes and the in vitro toxicity of $PM_{2.5}$ will specially be evaluated. Moreover, the sources of fine particulate matter will be determined based on the analysis of their nanostructure, morphology, and chemical composition. Finally, the social determinant assessments will be complemented with the driving forces methodology [46, 68, 69]. We will use Ⓡ and StataⓇ for the analysis.

## Patient and public involvement

Prior to start the study recruitment we will conduct focus groups with parents and caregivers to ensure, from their point of view, the best strategies for successful follow-up of children, and to validate that the data collection form is easy to understand. Likewise, we will develop the dissemination plan with parents and caregivers. We will present the results of the study from the baseline and follow-up, address the questions they might have, and then, we will work together to organize the information in a plain language to develop educational and recreational multimedia material. We will also invite kids included in the study and their families to co-creative sessions, where we will create some of the drawings, videos and other educational resources to use for dissemination of the study results.

On the other hand, representatives from the health care authorities have been participating as coinvestigators from the design, to identify the major needs and gaps in public health about air pollution and children under 5 years. In addition, two representatives of healthcare insurance company have been co-developed with the research team the recruitment and follow-up strategy to optimize the time of parents/caregivers and their kids during the execution of the study.

Finally, we will present the study results prior to the submission of the articles to parents and caregivers, then we will ask them if they want to participate in the interpretation of the study results and be co-authors of the articles, and/or whether they prefer to be in the acknowledgments section of all the material generated by the project.

## Data management

A Data Coordinating Center (DCC) will develop and manage the information system to centralize clinical information, as well as biological and environmental sample analysis results. The research team will guarantee that standard operating procedures and quality management activities such as standardized training of field and laboratory personnel in each city, and that ongoing verification of information quality and consistency are followed at all times. The DCC will report quality control progress in the documentation, laboratory results, biological and environmental samples, databases, and personnel training, every three months. The information collected will be encrypted and backed up weekly, and its access will be restricted to authorized researchers only.

## Discussion

This research study uses the exposome approach to characterize environmental pollutant exposure and its clinical respiratory and early biological effects on children under five, in two main Colombian cities. This is an unprecedented study conducted in South America that aims to provide evidence on the following aspects:

1. To estimate the toxic effect of particulate matter, transcending the approach based on pollutant concentration levels;

2. To estimate the risk of developing an upper and lower ARI, based on different exposure windows, including prenatal exposure;

3. To determine a baseline of early biological damage in children under five, and describe its progression after a one-year follow-up;

4. To describe how physical and chemical $PM_{2.5}$ characteristics influence toxicity and children's health;

5. To identify general external exposures (i.e., social determinants) associated with ARI.

Using the comparison approach to evaluate air quality and its effects on population health from a different perspective, has not been explored in South America. This research study will use alternative approaches to assess air pollution risks, such as the use of biomarkers that measure exposure to carcinogens like PAHs derived from engine combustion, as well as biomarkers that measure early DNA damage. Likewise, a clinical assessment of asthma will be conducted and complemented with an environmental *in vitro* risk assessment in cell lines. Furthermore, the chemical characterization of particulate matter, its shape and size, and their interaction with other environmental pollutants, i.e., meteorology, traffic conditions, and built spaces, will be jointly explored in our region.

This study aims to determine the synergistic effect that these environmental pollutants have on the onset and development of upper and lower ARI and asthma as the main childhood respiratory diseases in the world. In addition, it will provide information that will facilitate more in-depth studies of the association between air pollution sources and infant respiratory morbidity. In the end, this will possibly foster improvements in air quality control policies focused on toxicity levels and pollution sources, such as transportation, industry, smoking, and agricultural burning practices. Finally, this study will also provide information on population and environmental toxicity levels in a middle-income country, which may then be compared with the already available information of high-income countries.

Our study will have some limitations: measurements of indoor exposures will be based on questionnaires and we will not use samplers to determine the concentration of indoor contaminants due to budget constraints. We will not perform spirometry in children, and we do not have enough statistical power to find associations of the external exposome with asthma.

## Supporting information

**S1 File. Questionnaires used in the PROMESA study.**
(RAR)

## Acknowledgments

The authors would like to express their gratitude to the Medellín Higher Education Agency (SAPIENCIA) and Universidad Pontificia Bolivariana Agreement No. CDG19 for the DM PhD scholarship.

## Author Contributions

**Conceptualization:** Diana Marín, Luz Yaneth Orozco, Isabel Cristina Ortiz-Trujillo, María Victoria Toro, Zulma Vanessa Rueda.

**Funding acquisition:** Diana Marín, Juan José Builes.

**Investigation:** Luz Yaneth Orozco, Diana María Narváez, Carlos Daniel Ramos.

**Methodology:** Diana Marín, Luz Yaneth Orozco, Diana María Narváez, Francisco José Molina, Carlos Daniel Ramos, Laura Rodriguez-Villamizar, Shrikant I. Bangdiwala, Olga Morales, Martha Cuellar, Andrea Corredor, María Victoria Toro, Diego Alejandro Muñoz, Ferney Amaya, Ana Isabel Oviedo, Lucelly López, Beatriz Elena Marín-Ochoa, José Miguel Abad, Juan José Builes, Zulma Vanessa Rueda.

**Supervision:** Diana Marín, Diana María Narváez, Isabel Cristina Ortiz-Trujillo, Francisco José Molina, Olga Morales, Diego Alejandro Muñoz, Zulma Vanessa Rueda.

**Writing – original draft:** Diana Marín, Shrikant I. Bangdiwala, Juan José Builes, Zulma Vanessa Rueda.

**Writing – review & editing:** Diana Marín, Luz Yaneth Orozco, Diana María Narváez, Isabel Cristina Ortiz-Trujillo, Francisco José Molina, Carlos Daniel Ramos, Laura Rodriguez-Villamizar, Shrikant I. Bangdiwala, Olga Morales, Martha Cuellar, Luis Jorge Hernández, Enrique Antonio Henao, Verónica Lopera, Andrea Corredor, María Victoria Toro, Helena Groot, Milena Villamil-Osorio, Diego Alejandro Muñoz, Roberto Carlos Hincapié, Ferney Amaya, Ana Isabel Oviedo, Lucelly López, Ricardo Morales-Betancourt, Beatriz Elena Marín-Ochoa, Oscar Eduardo Sánchez-García, Juan Sebastián Marín, José Miguel Abad, Julio Cesar Toro, Eliana Pinzón, Juan José Builes, Zulma Vanessa Rueda.

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
