## [Decision Letter · Decision Letter 0]

22 Nov 2021

PONE-D-21-07992Characterization of the External Exposome and its Contribution to the Clinical Respiratory and Early Biological Effects in Children: The PROMESA Cohort Study ProtocolPLOS ONE

Dear Dr. Diana Marín,

Thank you for submitting your manuscript to PLOS ONE. After careful consideration, we feel that it has merit but does not fully meet PLOS ONE’s publication criteria as it currently stands. Therefore, we invite you to submit a revised version of the manuscript that addresses the points raised during the review process.

This is written article, but it needs minor revision following review by reviewers. 

The review comments can be found at the end of this email, together with any comments from the Editorial Office regarding formatting changes or additional information required to meet the journal’s policies at this time.

Please note that your revision may be subject to further review and that this initial decision does not guarantee acceptance.

We look forward to receiving your revised manuscript.

Kind regards,

Sajid Bashir Soofi

Academic Editor

PLOS ONE

2.  Please include additional information regarding the survey or questionnaire used in the study and ensure that you have provided sufficient details that others could replicate the analyses. For instance, if you developed a questionnaire as part of this study and it is not under a copyright more restrictive than CC-BY, please include a copy, in both the original language and English, as Supporting Information. Moreover, please include more details on how the questionnaire was pre-tested, and whether it was validated.

“JCT is the owner and CEO of ATB SERVICE SAS. The other authors do not have conflicts of interest to declare”

4. Thank you for stating the following in the Competing Interests/Financial Disclosure* (delete as necessary) section:

“DM, principal investigator. Grant number 121080763273, Contract 757-2018. This work was supported by the Colombian Ministry of Science, Technology and Innovation (https://minciencias.gov.co/). The funders had no role in study design, data collection and analysis, decision to publish, or preparation of the manuscript”

We note that one or more of the authors are employed by a commercial company: “Healthcare Company,  SURA.”

7. Your ethics statement should only appear in the Methods section of your manuscript. If your ethics statement is written in any section besides the Methods, please move it to the Methods section and delete it from any other section. Please ensure that your ethics statement is included in your manuscript, as the ethics statement entered into the online submission form will not be published alongside your manuscript.

8. We note that figure 1 in your submission contain map images which may be copyrighted. All PLOS content is published under the Creative Commons Attribution License (CC BY 4.0), which means that the manuscript, images, and Supporting Information files will be freely available online, and any third party is permitted to access, download, copy, distribute, and use these materials in any way, even commercially, with proper attribution. For these reasons, we cannot publish previously copyrighted maps or satellite images created using proprietary data, such as Google software (Google Maps, Street View, and Earth). For more information, see our copyright guidelines: http://journals.plos.org/plosone/s/licenses-and-copyright.

   1. You may seek permission from the original copyright holder of Figure(s) [#] to publish the content specifically under the CC BY 4.0 license. 

Reviewers' comments:

Reviewer's Responses to Questions

**Comments to the Author**

1. Does the manuscript provide a valid rationale for the proposed study, with clearly identified and justified research questions?

Reviewer #1: Partly

Reviewer #2: Yes

2. Is the protocol technically sound and planned in a manner that will lead to a meaningful outcome and allow testing the stated hypotheses?

Reviewer #1: Yes

Reviewer #2: Yes

3. Is the methodology feasible and described in sufficient detail to allow the work to be replicable?

Reviewer #1: Yes

Reviewer #2: Yes

4. Have the authors described where all data underlying the findings will be made available when the study is complete?

Reviewer #1: No

Reviewer #2: Yes

5. Is the manuscript presented in an intelligible fashion and written in standard English?

Reviewer #1: Yes

Reviewer #2: Yes

6. Review Comments to the Author

You may also provide optional suggestions and comments to authors that they might find helpful in planning their study.

Reviewer #1: The authors submitted here a research protocol for a study linking childhood exposome to respiratory health whereby the external exposome includes features of the built environment with a particular focus on air pollution while the internal exposome includes biomarkers associated with components of air pollution.

A point by point review is attached below.

Reviewer #2: This is well written manuscript. Here are few comments to consider in revised manuscript.

1. In abstract, expected study outcomes/findings should be added.

2. In introduction section, i suggest to replace 2017 GBD report with the recent GBD report.

3. In table 2, abbreviation of "t" should be added in footnotes.

7. PLOS authors have the option to publish the peer review history of their article (what does this mean?). If published, this will include your full peer review and any attached files.

Reviewer #1: No

Reviewer #2: No

---

## [Author Response · Author response to Decision Letter 0]

25 Jan 2022

January 25, 2022

Dear

Dr. Sajid Bashir Soofi

Academic Editor

Plos One

Thanks so much to both reviewers for their suggestions. All of them were very useful to improve our paper. Please find below the answer to each of their suggestions.

Reviewer 1

1. The authors submitted here a research protocol for a study linking childhood exposome to respiratory health whereby the external exposome includes features of the built environment with a particular focus on air pollution while the internal exposome includes biomarkers associated with components of air pollution.

A point by point review is attached below.

Answer: Dear reviewer thanks so much for all your comments. All of them were helpful to improve our paper.

Abstract

2. Specify if this work is focused on ambient (outdoor) vs. indoor exposures. 

Answer: Yes, we will evaluate both. We included it in the abstract 

3. Specify if asthma diagnosis is physician-based and if standardized protocol is used

Answer: We included in the abstract that the asthma diagnosis will be pediatric pulmonologist-based and a standardized protocol will be used. 

Introduction

4. Generally, the references used are not focused on the population of interest. For example, it would be a better fit to provide the global burden of disease among pediatric population. 

- Health effects references line 89 are mostly for the adult population, suggest revising the used references

- I would referring to work on genetic clock, telomere lengths and air pollution in early life. For example 

Isaevska E, Moccia C, Asta F, Cibella F, Gagliardi L, Ronfani L, Rusconi F, Stazi MA, Richiardi L. Exposure to ambient air pollution in the first 1000 days of life and alterations in the DNA methylome and telomere length in children: A systematic review. Environmental research. 2020 Nov 20:110504.

de Prado-Bert P, Ruiz-Arenas C, Vives-Usano M, Andrusaityte S, Cadiou S, Carracedo Á, Casas M, Chatzi L, Dadvand P, González JR, Grazuleviciene R. The early-life exposome and epigenetic age acceleration in children. Environment International. 2021 Oct 1;155:106683.

Answer: Thanks so much for your suggestions. We included information related to the global burden disease among pediatric population, the recommended references and additional references related to pediatric population. 

Material and Methods

5. It is debatable whether this is a prospective cohort or a retrospective birth cohort of the first 5 years of life. I would argue that it is the latter given the assessment of exposure prenatally and that the prospective work is only for 1 year. 

Answer: Your point is interesting. We think this is a prospective cohort study. The recruitment of children and the collection of information on environmental and indoor exposures, the outcomes and covariates of the study are prospective. 

We will take history of exposure about the children's diet during the last year and prenatal, specifically the parents' consumption of conventional cigarettes, electric cigarettes, smoked and inhaled drugs and the exposure to pollutants PM2.5 and NO2 during pregnancy. LUR models will be used for prenatal exposure to these two pollutants, and the history of exposure during pregnancy will be collected from the mother using the HELIX study questionnaire and an interview with the pediatric pulmonologist. Some variables collected during the interview with the mother will have cross verification with the clinical chart of the mother and children: prenatal control and perinatal information. 

All information will be collected at baseline, and in months 6 and 12.

6. More details are needed re: Health Insurance Company. Is this a national health coverage system? If not, would this requirement for the participants’ selection lead to biases? Is Colombia’s health care system a two tier system? More generally what does this criterion imply?

Answer: The Colombian health care system have two main ways of health insurance: “Contributive” means people who have work or have the payment capacity to contribute with a percentage of their monthly wage to healthcare access, or have special systems of health like teachers or military forces. “Subsidized” means for people who do not have employment, and cannot afford to pay health insurance, they are subsidized both from the government (state and local governments) as well as from resources from the contributive system. The benefits and access to health care for Contributive and Subsidized are the same. In 2020 there were 97.78% people affiliated to Contributive and Subsidized regimen in Colombia. 

The last category is “Not insured” for those that do not belong to any of the previous systems and need to pay with their own resources most of their health care (if the person is “Not insured” and has an emergency, the costs is paid by the state government). 

In addition, there are two structures: i) Health care insurance companies (can have clients from contributive and subsidized regimen) and, ii) Health care institutions that provide the clinical service. Sura health insurance company has clients from both Contributive and Subsidized regimen. 

In 2021, 94.3% of the population of Medellín was affiliated to SURA (health insurance company), and Bogotá 6.4%. We do not believe that there is a selection bias in Medellín. However, we believe that there could be a selection bias in Bogotá given the percentage coverage of SURA. We think this might be the selection biases and how we will address them:

1. Children from SURA could have different exposure patterns compared to those affiliated with other Health insurance companies and this could bias the studied associations. For example, exposure to air pollutants, tobacco smoke, eating patterns and other environmental conditions that have been associated with asthma and acute respiratory infections are related to some socioeconomic conditions. We will collect some variables about socioeconomic conditions for all children: Contributive or Subsidized, and strata (Colombia classified people according to their income in 6 categories, stratum 1 being the lowest income, and 6 the highest income).

The coverage of SURA affiliates to both schemes (Contributive and Subsidized) is similar to that of the general population, but there is a tendency to have less of the subsidized scheme (8% vs 17%). This could potentially bias the results towards the no-association hypothesis, but we believe that this bias could be mitigated because we are taking two sampling sites within each city whose social and environmental conditions are different, but similarly affect the children who live within each site. These sites were independently selected from the insurance company. 

We will evaluate in both cities the potential bias that the insurance regime could have in the distribution of the evaluated exposures and in the associations studied with the main exposures that we mentioned at the beginning.

2. The incidence of asthma and Acute Respiratory Infections (ARI) in SURA's children is different from those of other Health insurance companies. As we describe in the article, we selected the two sites within each city based on several criteria, none of them, related to the health insurance company nor the Contributive or Subsidized regimen. Therefore, we do not think this potential bias will be present in our study because children under 5 years of age are the second population group most affected by acute lower respiratory infections (ALRI) in Bogotá, and the two sampling sites rank second and third in areas that reported most cases of ALRI in the city of Bogotá (Document in Spanish available at http: //www.saludcapital.gov.co/DSP/Boletines%20sistemticos/Informe%20Anual/INFORME%20VIGILANCIA%20EN%20SALUD%20P%C3%9ABLICA%20BOGOT%C3%81%20A%C3%91O%20EPIDEMIOL%C3% 93GICO%202019%20-%2020200726.pdf).

We will evaluate the presence of this bias at the end of the study comparing the ALRI incidences of i) children included in the study, ii) children not included and who are affiliated with SURA, and iii) the general population. As at this time we do not know if there will be an overestimation or underestimation of the incidence, we will take into account the magnitude of this potential bias to adjust and discuss the results derived from the study.

3. Some confounding variables of the study associations vary according to the Health Insurance Company and this could bias the results. One of these potential confounders is the vaccination coverage of vaccine-preventable diseases for ALRI, which, if assumed to be different in SURA, would limit the inference of the results to the general population. However, Colombia historically has a high vaccination coverage in children under 5 years of age, between 93% to 96%. During COVID-19 pandemic those coverage decrease to 80% to 90% in several places. SURA has vaccination coverage of 95% until 2019, and 96.8% in 2020, similar to the tendency in Colombia and other health insurance companies. 

We will evaluate the effect of this potential confounder through a sensitivity analysis. Another possible confounder would be the history of prenatal control. The historical coverage of SURA has been 94%-97%, and in 2020 was 94.8%. This indicator in Colombia is high regardless of the insurer and we do not believe that it affects the results of the study.

In general, we will assess the impact of potential biases through sensitivity analysis and available analytical approaches by the time we finish collecting the information. Additionally, we emphasize that the participants will be studied regardless of their health condition or their level of exposure, but if the effect of potential biases is found, these will be acknowledge in the limitations of the study.

7. Line 196: what is the definition of prematurity? How is gestational age assessed?

Answer: infants born alive before 37 weeks of gestation. Gestational age is corroborated through three sources of information: 1) accessing the child's medical history when the growth and development program starts within the first eight days of birth, 2) asking the mother during the screening visit to verify compliance of the inclusion criteria, and 3) evaluation of clinical history by pediatric pulmonologist.

8. I would expect that only singleton births would be included – Why is this not the case here? 

Answer: Thank so much. We didn’t consider that criterion, as those kinds of pregnancies are not very common in Colombia. However, based on your question we will include it and only singleton births would be included. 

Objectives and Outcomes Measurements

9. A major limitation is the fact that asthma diagnosis at age 5 is usually complex. The authors have provided a national guideline document, but (1) it is in Spanish; (2) it is unclear if this is valid for age 5 years.

Answer: To establish the diagnosis of asthma in our study, we follow the “Clinical Practice Guide for the diagnosis, comprehensive care and follow-up of children with a diagnosis of Asthma”, which is a Colombian guide based on the best existing international information about the prevalence, clinical course, diagnosis, non-pharmacological treatment (primary prophylaxis, secondary prophylaxis), pharmacological treatment (chronic therapy, treatment of attacks or exacerbations), follow-up and prognosis of asthma in pediatric patients including preschool children (<5 years).

This guide establishes that, in children under 5 years of age, with a history of wheezing, cough, respiratory distress syndrome and chest tightness, asthma is more likely if the symptoms are:

- They are frequent and recurrent;

- Vary with time and intensity of symptoms;

- Worse at night or early in the morning;

- Triggered or worsened by exercise, exposure to pets, cold or damp air, laughter, and emotions;

- They occur without the need for an accompanying infection of the upper respiratory tract;

- They occur when there is a personal or family history of atopy, asthma or both;

- They are accompanied by generalized wheezing on pulmonary auscultation;

The asthma predictor index will be applied to help identify preschoolers with recurrent wheezing at high risk of developing persistent asthma symptoms, as follows:

Primary criterion

Three or more episodes of wheezing in the past year

Secondary criteria

I. Major criteria

- Have a parent with asthma

- Have atopic dermatitis

II. Minor criteria

- Medical diagnosis of allergic rhinitis

- Wheezing not related to viral infections

- Peripheral eosinophilia equal to or greater than 4%

*It is considered positive if it presents the primary criterion, plus at least one major criterion or two minor criteria.

For each diagnosis of asthma, the severity will be classified according to the table.

Table. Classification of asthma according to its severity for children under 5 years of age. 

Asthma category Symptoms Nocturnal Symptoms Use of �2 for symptom control* Interference with normal activity Asthma attacks requiring oral corticosteroids**

Mild Intermittent ≤1 day per week None ≤2 per week None 0-1 per year

Mild persistent >2 days per week but no daily 1-2 times per month >2 days per week but not daily Minimum limitation ≥2 attacks requiring steroids in 6 months, or 4 or more wheezing episodes per year lasting more than 1 day, and risk factors for persistent asthma

Persistent moderate Daily symptoms 3-4 times per month Daily Some limitation ≥2 attacks requiring steroids in 6 months, or 4 or more wheezing episodes per year lasting more than 1 day, and risk factors for persistent asthma

Persistent severe Ongoing symptoms (throughout the day) Frequent Several times a day Extreme limitation ≥2 attacks requiring steroids in 6 months, or 4 or more wheezing episodes per year lasting more than 1 day, and risk factors for persistent asthma

*No prevention of symptoms with exercise. 

** Exacerbations, regardless of their severity, can occur in any category and their frequency can vary over time.

10. Line 225: given the evolution of the pandemic, do authors truly believe that the timeline for recruitment is feasible?

Answer: Given the conditions of the pandemic, a change had to be made in the schedule of the study, which involved requesting an extension from the financing entity (Minciencias). The project will be conducted until January 2024 and we updated the article accordingly.

Laboratory Evaluations, Assays, and Handling

11. What type of selection bias would the authors expect from the selection of these list of genes?

Answer: Thanks for your question. There are three important points to consider: 1) The children will not be selected based on the genes, 2) The genes will be processed at the end of the recruitment, even at the end of the follow-up, and 3) The criteria for selecting these genes were: 

- We have budget to process only five genes. If we get additional funding, we will expand the list.

- To select the five genes, we prioritized them based on biological plausibility based on published literature about the association between exposure to Polycyclic Aromatic Hydrocarbons (PAHs), which are the main contaminants identified in PM2.5, and risk of cell damage or cell toxicity, as well as genes involved in signaling and repair processes. Studies have shown that polymorphisms in CYP1A1, Glutathione S-Transferase and Aryl Hydrocarbon Receptor are linked to increased biomarkers of exposure and damage, such as micronuclei, repair, and genotoxicity from exposure to air pollutants. There is also increasing evidence that reflects the association of polymorphisms in H2AX and P53 genes with deficiencies in the recognition of repair processes and therefore greater mutation and damage associated with exposure to atmospheric pollutants.

- We will continue to read new emerging evidence during the execution of the project and until the moment we will process the samples to evaluate those polymorphisms. If there is an update in the evidence, we will reconsider which genes would be the most appropriate to evaluate the susceptibility of children. 

We recognize that some of the possible biases that could occur with these genes would be:

1. A potential information bias could occur by not evaluating the epigenetic modifications and gene expression. We will not know if the polymorphisms in these genes are associated with their expression. 

2. Genetic modifications are timeless and this means that long-term effects cannot be evidenced and associated with these biomarkers. More extensive follow-up is required to achieve the clinical outcomes of interest and obtain more robust associations.

3. Biomarkers may be related to multigene interactions that are associated with environmental factors and we will not evaluate the gene networks involved with these biomarkers.

We acknowledge these potential limitations, but our study is a starting project that has limited funding. We will store the samples to be able to answer future research questions when we are able to expand our funding.

12. Lines 268-271: how will the sampling sites be chosen? What risks associated with high volume samples over 2 weeks in heavily polluted areas and conversely with low volume samplers in "cleaner" areas?

Answer: This is a very important question. Lines 181-194 of the article describes the criteria that we used to select the sampling sites. 

Our sampling sites were selected based on the following criteria:

1. Three different sources of air quality measurements identified high and low PM2.5 levels in the same areas in 2017, 2018, and 2019:

a. High Concentration Zones: inventory of emissions above the 90th percentile, air quality monitoring stations with the highest number of days with a PM2.5 Air Quality Index between 101 and 200. For these zones, the WRF, BRAMS-CAMx simulation model yields values exceeding 37 �g/m3.

b. Low Concentration Zones: inventory of emissions below the 10th percentile, air quality monitoring stations with the highest number of days with a PM2.5 Air Quality Index between 0 and 50. For these zones, the WRF, BRAMS-CAMx simulation model yields values below 15 �g/m3.

2. At least one healthcare provider from the Health Insurance Company must be operating within these high and low concentration zones.

3. At least one Air Quality Monitoring Station is operating within each zone.

Regarding your second question, the ideal would have been to carry out the sampling in parallel at the two sampling sites, however, this was not possible due to budget limitations. To fix this, we will select the same days of the week to sample and the same number of samples at the sampling sites. Medellin and Bogota have previous research from other researchers that show that the emission conditions do not vary that much within the month. It has also been identified the months of higher and lower concentrations of PM2.5. 

We carried out pilot tests in both sampling sites to determine what would be the minimum concentration of PM2.5 to be able to carry out the required analyzes. We did not find significant variations in the volumes at each sampling site and the sampling methodology that we will use to capture the PM2.5, as well as the number of samples taken are enough to meet our objectives.

Our sampling methodology are using Hi Vol and Low Vol, both are approved by the EPA. However, Hi Vol sampling is generally considered more representative, since the volume of filtered air is greater and therefore the mass collected is greater than Low Vol sampling, which improves accuracy of gravimetric method. Therefore, by having two methodologies (Hi Vol and Low Vol) performing parallel sampling, the results are more robust for periods of low and high concentrations of PM2.5.

Sample Size Selection and Statistical Analyses

13. RR chosen by the authors is based on the ESCAPE project publications, this is for upper ARI based and for likely lower exposure levels that what would be expected in the study sites of the proposed research protocol. Therefore, I would expect a much lower effect estimate of lower ARI and asthma. Various simulations (high, medium, and low exposure levels along with high and low RR) would allow a range of sample size to consider. 

Answer: Thank you for the comment. The calculation of the sample size was not adequately explained. 

Several clarifications:

1. We estimated the sample size for ALRI.

2. We do not have data in Colombia about the estimation of the incidence of ALRI in low air pollution zone.

3. We used the data from the RR of ESCAPE as a reference to estimate how much the incidence of ALRI could be in the low air pollution zone. They reported a combined odds ratio (OR) of Pneumonia and PM2.5 of 2.58 (95% CI 0.91 – 7.27). 

4. In addition, a cohort study in Colombia (reference 14) showed that the incidence of ALRI in a high air pollution zone was 37.5%. 

5. Then the incidence in areas of low air pollution, assuming that it was similar to ESCAPE, would be 14.5% (Incidence of low air pollution = 37.5/2.58). With these parameters, the estimated sample size would be 128 children. 

6. We know that in Colombia the levels of contamination can be higher as well as the incidences of ALRI. 

7. To estimate our sample size reported in the article, we used an RR of 1.48. We projected a much lower estimator than the one of ESCAPE, as we thought exactly the same as the reviewer. With an RR of 1.48 our sample size was 500 children.

A recent meta-analysis by Wang X et al. on the global burden of respiratory infections in <5 years shows that the incidence (per 1000 children per year) of Human parainfluenza virus (hPIV)-associated ALRI cases is around 37.8% in Low-income, lower-middle-income countries, and high-income countries which supports our calculations. 

Additionally, as shown by this meta-analysis and public health reports in Colombia, the expected number of cases of ALRI is much higher in low- and middle-income countries compared to high-income countries, and the peaks of ALRI cases occur in the months of May-June and September-November, just after the air pollution peaks occur. These months also have higher levels of rainfall.

Finally, regarding asthma, objective 3 is an exploratory objective because we recognize that the sample size does not have enough power to find robust associations. We mentioned this point in the article.

14. Line 332: how are the authors planning to measure surrounding natural spaces?

Answer: We will use two sources of information: the questionnaire and geographic information

system-based models and we will use the Normalized Difference Vegetation Index.

15. Lines 341 to 343: this is quite vague and presents different approaches with little information on how the modeling will be conducted. 

Answer: Thanks for your comment. We apologize because we keep it short due to the extension of the paper. Until now, there no consensus that establishes which statistical methods should researchers do in exposome-health association studies. However, given the variety in the types of outcomes and exposures (numerical and categorical) in our project, there are several methods that we could use to deal with multiple testing, false positives and false-negative results, confounding and interaction. Some methods that show good performance in dealing with these challenges are Deletion–Substitution–Addition (DSA) algorithm, Bayesian Kernel Machine regression, and Weighted Quantile Sum Regression. Additionally, another machine learning method that addresses repeated measures analysis is Elastic Net (ENET) and Least Absolute Shrinkage and Selection Operator (LASSO). 

We included this information in the statistical analysis.

Discussion

16. One of the major limitations of this work is the contribution of the indoor exposures and the fact that all exposures are based on residential home address. This should be addressed/commented on in the discussion

Answer: Thanks for your comment. We consider the following aspects:

1. The measurement of environmental exposures contemplates the use of EPA-approved methodologies for the capture of PM2.5 that will allow us to characterize the sampling site. We will use GIS-based questionnaires and models to identify exposure to traffic and green spaces. The questionnaire investigates the exposure in the place where the children study and the green spaces where he plays within the city.

2. Not all environmental exposures are based on the address of the participant, only those that correspond to the LUR models of PM2.5 and NO2. However, the fact that some of the exposures are based on the address of residence rather than being a limitation is an advantage because of the accuracy in the measurement of the exposure. We will use GIS to improve the estimation of some of the external exposures.

3. Indoor exposures will be evaluated through questionnaires. It is true that we do not have a portable device that children can carry with them and “quantitative” indoor exposure. We will recognize this limitation in the discussion. 

Reviewer 2

1. This is well written manuscript. Here are few comments to consider in revised manuscript. In abstract, expected study outcomes/findings should be added.

Answer: Thanks for your comment. We included your suggestion in the abstract.

2. In introduction section, I suggest to replace 2017 GBD report with the recent GBD report.

Answer: Thanks for the suggestion. It was replaced by 2019 GBD.

3. In table 2, abbreviation of "t" should be added in footnotes.

Answer: We explained the abbreviation of “t”.

Thank you for receiving our manuscript and considering it for review. We appreciate your time and look forward to hearing from you at your earliest convenience. 

Sincerely,

Diana Marín

On behalf of the authors

---

## [Decision Letter · Decision Letter 1]

28 Nov 2022

Characterization of the External Exposome and its Contribution to the Clinical Respiratory and Early Biological Effects in Children: The PROMESA Cohort Study Protocol

PONE-D-21-07992R1

Dear Dr. Diana Marín,

We’re pleased to inform you that your manuscript has been judged scientifically suitable for publication and will be formally accepted for publication once it meets all outstanding technical requirements.

Within one week, you’ll receive an e-mail detailing the required amendments. When these have been addressed, you’ll receive a formal acceptance letter, and your manuscript will be scheduled for publication.

An invoice for payment will follow shortly after the formal acceptance. To ensure an efficient process, please log into Editorial Manager at http://www.editorialmanager.com/pone/, click the 'Update My Information' link at the top of the page, and double check that your user information is up to date. If you have any billing related questions, please contact our Author Billing department directly at authorbilling@plos.org.

Kind regards,

Sajid Bashir Soofi

Academic Editor

PLOS ONE

Additional Editor Comments (optional):

Thanks for addressing the comments. Congratulations, your manuscript is accepted for publication.

Reviewers' comments:

Reviewer's Responses to Questions

**Comments to the Author**

1. Does the manuscript provide a valid rationale for the proposed study, with clearly identified and justified research questions?

Reviewer #3: Yes

2. Is the protocol technically sound and planned in a manner that will lead to a meaningful outcome and allow testing the stated hypotheses?

Reviewer #3: Partly

3. Is the methodology feasible and described in sufficient detail to allow the work to be replicable?

Reviewer #3: Yes

4. Have the authors described where all data underlying the findings will be made available when the study is complete?

Reviewer #3: Yes

5. Is the manuscript presented in an intelligible fashion and written in standard English?

Reviewer #3: Yes

6. Review Comments to the Author

You may also provide optional suggestions and comments to authors that they might find helpful in planning their study.

Reviewer #3: 1. Environmental pollution as been regarded as an important trigger factor for Bronchial asthma. The concept of external exposome causing various respiratory diseases in children has gotten a potential merit to be investigated.

not only the outdoor pollution but also indoor exposures also need to be evaluated. Making a diagnosis of asthma should be scientifically guided. And the study children should have a good family history and a good birth history.

7. PLOS authors have the option to publish the peer review history of their article (what does this mean?). If published, this will include your full peer review and any attached files.

Reviewer #3: **Yes: **Professor Khan Abul Kalam Azad

Assisted By Dr. S M Kawser Zafor Prince

---

## [Editor Report · Acceptance letter]

5 Jan 2023

PONE-D-21-07992R1 

Characterization of the External Exposome and its Contribution to the Clinical Respiratory and Early Biological Effects in Children: The PROMESA Cohort Study Protocol 

Dear Dr. Marín:

I'm pleased to inform you that your manuscript has been deemed suitable for publication in PLOS ONE. Congratulations! Your manuscript is now with our production department. 

Kind regards, 

on behalf of

Professor Sajid Bashir Soofi 

Academic Editor

PLOS ONE